# Protein Interactome Analysis of the Type IX Secretion System Identifies PorW as the Missing Link between the PorK/N Ring Complex and the Sov Translocon

Dhana G. Gorasia,[a] Ignacio Lunar Silva,[b] Catherine A. Butler,[a] Maïalène Chabalier,[b] Thierry Doan,[b] Eric Cascales,[b] Paul D. Veith,[a] Eric C. Reynolds[a]

aOral Health Cooperative Research Centre, Melbourne Dental School, Bio21 Institute, The University of Melbourne, Parkville, Victoria, Australia

bLaboratoire d'Ingénierie des Systémes Macromoléculaires (LISM - UMR7255), Institut de Microbiologie, Bioénergies et Biotechnologie (IM2B), CNRS - Aix-Marseille Universite, Marseille, France

Dhana G. Gorasia and Ignacio Lunar Silva contributed equally to this article. Author order was determined by order of increasing input in writing and experimental design.

**ABSTRACT** The type IX secretion system (T9SS) transports cargo proteins through the outer membrane of *Bacteroidetes* and attaches them to the cell surface for functions including pathogenesis, gliding motility, and degradation of carbon sources. The T9SS comprises at least 20 different proteins and includes several modules: the trans-envelope core module comprising the PorL/M motor and the PorK/N ring, the outer membrane Sov translocon, and the cell attachment complex. However, the spatial organization of these modules is unknown. We have characterized the protein interactome of the Sov translocon in *Porphyromonas gingivalis* and identified Sov-PorV-PorA as well as Sov-PorW-PorN-PorK to be novel networks. PorW also interacted with PGN_1783 (PorD), which was required for maximum secretion efficiency. The identification of PorW as the missing link completes a continuous interaction network from the PorL/M motor to the Sov translocon, providing a pathway for cargo delivery and energy transduction from the inner membrane to the secretion pore.

**IMPORTANCE** The T9SS is a newly identified protein secretion system of the *Fibrobacteres-Chlorobi-Bacteroidetes* superphylum used by pathogens associated with diseases of humans, fish, and poultry for the secretion and cell surface attachment of virulence factors. The T9SS comprises three known modules: (i) the trans-envelope core module comprising the PorL/M motor and the PorK/N ring, (ii) the outer membrane Sov translocon, and (iii) the cell surface attachment complex. The spatial organization and interaction of these modules have been a mystery. Here, we describe the protein interactome of the Sov translocon in the human pathogen *Porphyromonas gingivalis* and have identified PorW as the missing link which bridges PorN with Sov and so completes a continuous interaction network from the PorL/M motor to the Sov translocon, providing, for the first time, a pathway for cargo delivery and energy transduction from the inner membrane to the secretion pore.

**KEYWORDS** *Porpyromonas gingivalis*, T9SS, gingipains, secretion

Gram-negative bacteria have developed numerous ways to transport cargo proteins across the outer membrane (OM). To date, 10 different types of secretion system (T1SS to T10SS) have been identified in bacteria (1–5). The type IX secretion system (T9SS) is specific to the *Fibrobacteres-Chlorobi-Bacteroidetes* (FCB) superphylum and has been studied mainly in *Porphyromonas gingivalis*, found in dental plaque, and *Flavobacterium johnsoniae*, found in soil and freshwater (6–9). In *F. johnsoniae*, the T9SS secretes cell-surface adhesins that are required for gliding motility (3, 10–13), while in *P. gingivalis*, a human oral pathogen highly associated with periodontitis (14), the T9SS

Address correspondence to Eric Cascales, cascales@imm.cnrs.fr, Paul D. Veith, pdv@unimelb.edu.au, or Eric C. Reynolds, e.reynolds@unimelb.edu.au.

The authors declare no conflict of interest.

secretes cell-surface virulence factors, such as gingipains (3). The *P. gingivalis* T9SS secretes ~30 other cargo proteins (7, 8, 15, 16), including the carboxypeptidase CPG70 (17) and peptidylarginine deiminase (PPAD), an enzyme responsible for host protein citrullination that has been linked to rheumatoid arthritis (18).

The T9SS cargo proteins have an N-terminal signal peptide for export across the inner membrane by the Sec system and a conserved C-terminal domain, referred to as the CTD signal, that enable them to pass through the OM via the T9SS (15, 16, 19).

The T9SS is comprised of at least 20 component proteins, namely, PorK, PorL, PorM, PorN, Sov, PorT, PorU, PorW, PorP, PorV, PorQ, PorZ, PorE (PG1058), PorF (PG0534), PorG (PG0189), Plug (PG2092/PGN_0144), PorA (PG2172/PGN_0123), and three transcription regulators, PorX, PorY, and SigP (3, 20–27). *P. gingivalis* is black pigmented when grown on blood agar. This pigmentation is generally lost in the *P. gingivalis* T9SS mutants due to the loss of cell-surface-attached gingipains. The T9SS includes three modules: the trans-envelope complex, the translocon complex, and the attachment complex (7–9). The trans-envelope complex comprises two inner membrane proteins, PorL and PorM, and two proteins associated with the OM, PorN and PorK (3, 28, 29). Five PorL and two PorM subunits assemble a molecular motor that uses the proton-motive force to power substrate transport (30). The PorK lipoprotein and the PorN periplasmic protein form a double-layered ring located underneath the OM and comprising 32 to 36 subunits of each protein (27). Sov/SprA is a 36-$\beta$-strand barrel and constitutes the major protein of the translocon complex (25). Sov/SprA interacts with the Plug protein or the PorV $\beta$-barrel (25). In the Plug complex, the periplasmic gate of the translocon is closed; the Plug preventing the passage of the cargo proteins (25). In the PorV complex, the entrance is free, allowing cargo to transit through the translocon and bind to PorV at the cell surface (25). It is proposed that PorV then shuttles cargo from the translocon to the attachment complex (31). In addition to PorV, the attachment complex comprises PorU, PorZ, and PorQ (31). The PorU protease acts as a sortase (32) by cleaving the CTD signal and anchoring the processed cargo protein to a specific form of lipopolysaccharide (LPS) called anionic-LPS (A-LPS) (32–35). PorZ is a carbohydrate-binding protein that is suggested to supply A-LPS to the PorU sortase for cargo attachment (21, 36).

The remaining conserved components of the T9SS, including PorT, PorW, PorF, PorP, and PorE, are poorly characterized. Furthermore, while several functional complexes have been identified and characterized, the spatial organization of the three modules to each other is unknown. In this study, we sought to better understand how the translocon is connected to the PorKLMN trans-envelope complex. Here, for the first time, we provide evidence that the OM PorW lipoprotein is the missing link that forms a bridge between Sov and PorN.

## RESULTS

**PorW is required for Sov to form a high-molecular-weight complex.** To identify Sov-binding partners, Sov-containing complexes were examined by BN-PAGE immunoblot using Sov-specific antibodies. For this analysis, we used whole-cell lysate from the ABK⁻ strain, which lacks all three gingipains but has a functional T9SS. Elimination of gingipains in the sample minimizes protein degradation, and since gingipains are the most abundant proteins, their exclusion allows better detection of low abundance proteins. Two Sov-specific bands were detected in the immunoblot, one at a molecular weight of ~500 kDa and another of ~750 kDa (Fig. 1A). To investigate the composition of these bands, cell lysates of the *P. gingivalis porW*, *porV*, and *porP* mutants were analyzed. The band at ~500 kDa was detected in all the mutants; however, the band at ~750 kDa was detected in the *porV* and *porP* mutants but was missing in the *porW* mutant (Fig. 1A). To examine the presence of PorW at ~750 kDa and its native migration profile, we sliced the BN-PAGE gel lane containing proteins from the ABK⁻ strain into 12 protein bands (Fig. 1B) and subjected it to trypsin digestion and mass spectrometry (MS) analysis. We chose mass spectrometry because the PorW antibody we generated failed to recognize PorW in the native immunoblot. Figure 1C and D show the native

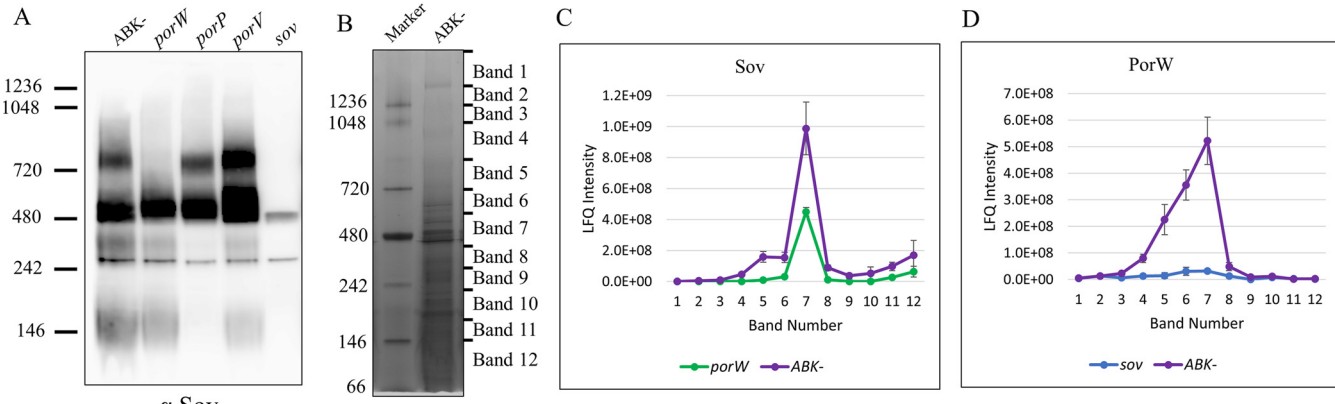

**FIG 1** PorW is required for Sov to form the 750-kDa complex. (A) The *P. gingivalis* ABK⁻ strain (expressing a functional T9SS) together with the T9SS mutants (*porW*, *porP*, *porV*, and *sov*) were lysed in 1% DDM and electrophoresed on a BN-PAGE gel. The proteins were transferred onto a PVDF membrane and probed with Sov-specific antibodies. The *sov* mutant served as a negative control for the nonspecific binding of the Sov-specific antibodies. (B) The *P. gingivalis* ABK⁻ cells were lysed in 1% DDM, electrophoresed on a BN-PAGE gel, and stained with Coomassie blue. The gel lane was sliced into 12 bands and an in-gel tryptic digestion was performed on the gel pieces. Tryptic peptides were analyzed by mass spectrometry and identified using MaxQuant software. The LFQ intensities of (C) Sov and (D) PorW in ABK⁻, *porW*, and *sov* mutants were plotted.

migration profiles of the Sov and PorW proteins, respectively. In the ABK⁻ strain, Sov appeared to have a small peak at band 5 and a large peak at band 7 consistent with the two bands in the immunoblot data. PorW appeared to have a broader peak from bands 4 to 7 in the ABK⁻ strain, possibly indicating its involvement in multiple interactions. The small Sov peak at gel band 5 disappeared in the *porW* mutant while the peak at gel band 7 remained (Fig. 1C), supporting the immunoblot result that the ~750 kDa Sov complex is PorW dependent. The peak of PorW protein flattened in the *sov* mutant, suggesting that its high-molecular-weight complexes are dependent on Sov (Fig. 1D). Collectively, the data suggest a close functional association and potential physical interaction between PorW and Sov.

**Sov associates with multiple components of the T9SS.** To identify additional Sov partners, we performed coimmunoprecipitation on the *P. gingivalis* ABK⁻ strain using Sov-specific antibodies. The immunoprecipitated material was quantified by mass spectrometry and MaxQuant relative to the *sov* mutant negative control. The label-free quantitation (LFQ) intensity ratios of the top 30 proteins were plotted (Fig. 2A). PorV and Plug were observed in the pulldown assay with high LFQ intensity ratios (Fig. 2A) consistent with their known interaction with Sov (25). PorW, along with PGN_1783 (designated PorD), PorA, PorK, PorN, PorZ, and PorE, also exhibited high LFQ intensity ratios (Fig. 2A), indicating that their presence in the pulldown assay was specific to the presence of Sov. PorA was recently demonstrated to be a component of the T9SS and is also known as a substrate of the T9SS (26). The specific enrichment of many T9SS components in the Sov coimmunoprecipitated material suggests a large T9SS interactome involving Sov. In addition to PorA, several other cargo proteins, such as PGN_0335 (CPG70), PGN_0659 (Hbp35), PGN_1115, PGN_1416 (PepK), and PGN_1556 (adhesin), also had high LFQ intensity ratios, consistent with their secretion through the Sov pore (Fig. 2A, cargo proteins indicated with an asterisk). To estimate the relative abundance of the proteins that coprecipitated with Sov, the iBAQ metric was used. After correction of the iBAQ values using the *sov* mutant negative control, PorV and PorA were the most abundant partners at 35% and 22% relative to the level of Sov, with all other interacting components being <4% (Fig. 2B, Table 1).

To investigate the linkage of these proteins to Sov and whether the interactions changed in T9SS mutant strains, a panel of mutants comprising *porW*, *porV*, *porN*, *porT*, and *porP* as well as a newly created mutant (*porD*, see below) were also subjected to Sov antibody immunoprecipitation, and the iBAQ metric was used to examine the changes in the relative abundance in the immunoprecipitated proteins. For all strains tested, the levels of PorV and PorA closely followed each other, and in the *porV* mutant,

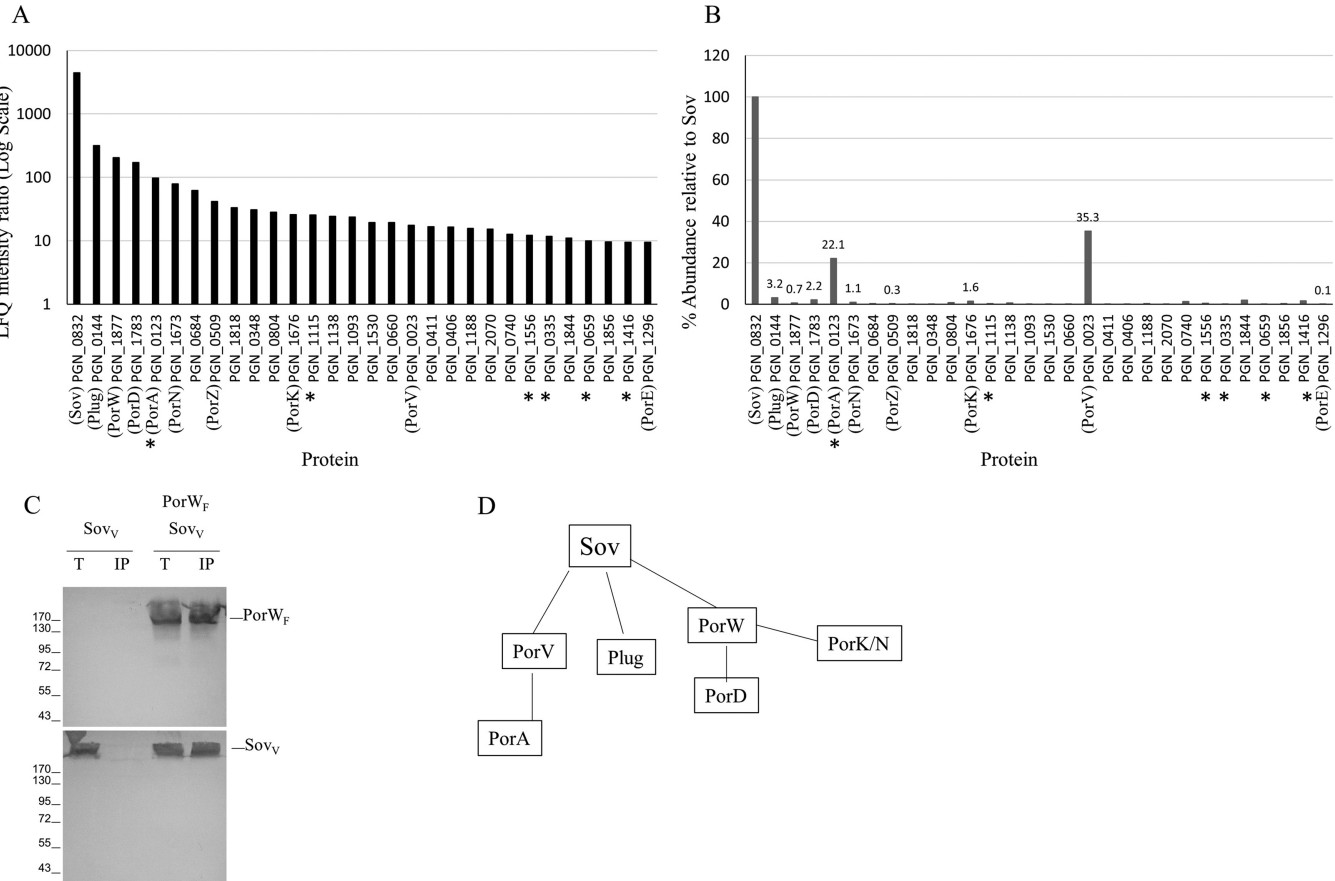

**FIG 2** Sov interacts with multiple components of the T9SS. The *P. gingivalis* ABK⁻ strain was subjected to coimmunoprecipitation using Sov-specific antibodies. The *sov* mutant was used as a negative control. The immunoprecipitated samples were digested with trypsin, analyzed by mass spectrometry, and quantified using MaxQuant software. (A) The ratio of LFQ intensities (ABK⁻/sov) of the top 30 proteins was plotted. T9SS cargo proteins are marked with an asterisk (*). (B) For the same 30 proteins, the percentage of abundance relative to Sov using iBAQ intensities was plotted. (C) Detergent-solubilized lysates of *E. coli* cells producing FLAG-tagged PorW (PorW_F) and VSV-G-tagged Sov (Sov_V) were subjected to immunoprecipitation with anti-FLAG-coupled beads. The total lysates (T) and immunoprecipitated (IP) material were separated by 10% SDS-PAGE and immunodetected with anti-FLAG (upper panels) and anti-VSV-G (lower panels) monoclonal antibodies. Molecular weight markers are indicated on the left. (D) Summary of the proposed Sov interactions. Note: to show the LFQ intensity ratio of Sov in panel A, the lowest LFQ intensity (of any protein) identified in the pulldown assay of the *sov* mutant was used instead of a zero.

PorA was completely absent, indicating that its interaction with Sov requires PorV (Table 1, Fig. 2D). PorV was present at up to 1:1 stoichiometry with Sov and was usually at least 10-fold more abundant than Plug, with the exceptions being the *porV* mutant where the stoichiometry of the Plug rose to 140% and the *porD* mutant where PorV and Plug were of similar abundance (Table 1).

For the *porW* mutant, all the major components identified in ABK⁻ were present except for PorD, suggesting that PorD is dependent on PorW for its interaction with Sov (Table 1). Additionally, PorK and PorN were substantially reduced in the *porW* mutant with LFQ ratios less than 10 and iBAQ percentage of <0.4, suggesting that they may also depend on PorW for their interaction with Sov (Table 1). In the *porN* mutant, the relative abundances of PorW and PorD were not adversely affected, suggesting that the PorW/PorD interaction with Sov was not dependent on PorN (Table 1). Similarly, in the *porD* mutant, the relative abundances of PorW, PorK, and PorN were not adversely affected, suggesting that the interaction of PorW/PorK/PorN with Sov was independent of PorD (Table 1). The simplest way to interpret these dependencies is through an interactome where PorW binds directly to Sov while PorD and the PorK/N ring form independent connections to PorW (Fig. 2D).

**PorW binds directly to Sov.** Sov is thus involved in numerous interactions, including with PorV, Plug, and PorW. The Sov-PorV and Sov-Plug complexes have already been characterized (25). To test whether Sov and PorW interact directly, we performed

**TABLE 1** Relative abundance of Sov-binding partners in several T9SS mutants[a]

| Protein name | LFQ ratio ABK⁻ | iBAQ % for: | | | | | | |
|---|---|---|---|---|---|---|---|---|
| | | ABK⁻ | porP | porT | porW | porV | porN | porD |
| Sov | 4,472 | 100.0 | 100.0 | 100.0 | 100.0 | 100.0 | 100.0 | 100.0 |
| PorV | 18 | 35.3 | 77.4 | 104.4 | 93.6 | 0.0 | 69.3 | 20.9 |
| PorA | 98 | 22.1 | 69.2 | 99.7 | 60.4 | 0.0 | 48.4 | 16.7 |
| Plug | 319 | 3.3 | 6.9 | 1.5 | 1.4 | 142.1 | 0.1 | 17.7 |
| PorW | 205 | 0.7 | 2.7 | 1.3 | 0.0 | 1.9 | 4.5 | 3.0 |
| PorD | 172 | 2.2 | 5.2 | 4.5 | 0.0 | 7.9 | 10.6 | 0.0 |
| PorK | 26 | 1.6 | 2.2 | 4.5 | 0.4[b] | 5.2 | 0.0 | 1.3 |
| PorN | 80 | 1.1 | 0.7 | 2.8 | 0.1[b] | 2.5 | 0.0 | 2.1 |
| PorZ | 42 | 0.3 | 0.0 | 0.0 | 0.0 | 0.0 | 0.0 | 0.5 |
| PorE | 10 | 0.1 | 0.0 | 0.0 | 0.0 | 0.0 | 0.1 | 0.3 |

[a]iBAQ % shaded in gray had LFQ ratios of ≥10 and were deemed highly significant.
[b]PorK (LFQ ratio = 3), PorN (LFQ ratio = 6) were not considered significant.

pulldown experiments in the heterologous host *Escherichia coli* (i.e., devoid of T9SS components) producing FLAG-tagged PorW (PorW$_F$) and VSV-G tagged Sov (Sov$_v$) using anti-FLAG-coupled beads. Pulldown experiments using detergent-solubilized membrane fractions showed that PorW coprecipitated with Sov (Fig. 2C), confirming a direct interaction between the two proteins.

**The C-terminal domain of PorW interacts with PorN.** Since the immunoprecipitation of PorK and PorN with Sov in *P. gingivalis* was PorW dependent, we explored whether these proteins, as well as the PorL and PorM trans-envelope proteins, interacted with PorW using a bacterial two-hybrid system (BACTH). First, the *porW* gene devoid of the sequence coding for its putative secretion signal and N-terminal cysteine (see below) was cloned into bacterial two-hybrid vectors. BACTH assays showed that PorW interacted with PorN, but no interaction was detected with PorK, the cytoplasmic domain of PorL, or the periplasmic domain of PorM (Fig. 3A). Pulldown experiments on anti-FLAG affinity agarose gel confirmed that FLAG-tagged PorW coprecipitated with VSV-G-tagged PorN (Fig. 3B) from detergent-solubilized membrane fractions.

*In silico* Pfam and TPRPred analyses of the PorW protein identified two arrays of tetra-tricopeptide repeat (TPR) motifs (Fig. S1). In *P. gingivalis*, the TPR1 and TPR2 domains are predicted to comprise 9 and 2 consecutive TPR motifs, respectively. These TPR motifs are generally conserved in PorW/SprE proteins from different species (Fig. S1), suggesting a functional role. We tested the contribution of PorW TPR1 (amino acids [aa] 125 to 400 [numbering of the mature form]) and TPR2 (aa 589 to 684) motifs to the interaction with PorN. Using BACTH, we did not detect TPR1-PorN or TPR2-PorN interactions. We thus predicted the PorW structure using Phyre2 and AlphaFold2. The predicted structure showed that PorW is an $\alpha$-helical protein (Fig. 3C). We segmented PorW into 6 subdomains, based on helix packing and the location of potential flexible linkers (D1 to D6 constructs; D1, aa 1 to 94; D2, aa 95 to 258; D3, aa 259 to 388; D4, aa 389 to 788; D5, aa 789 to 1,051; D6, aa 1,052 to 1,161) plus a D7 construct comprising D2 to D4 (aa 95 to 788) and encompassing both TPR1 and TPR2 (Fig. 3C). BACTH assays showed that PorN interacted with the PorW D6 C-terminal domain (Fig. 3D), a result that was further confirmed by pulldown assay (Fig. 3E). However, we noticed that the PorW-D6 interaction with PorN was more labile than the PorW-PorN complex, as it needed to be stabilized by chemical cross-linking with *para*-formaldehyde, suggesting that additional regions of PorW are required to stabilize PorW-PorN interactions. Therefore, we conclude that PorW forms a direct bridge between the PorK/N rings and the Sov translocon.

**PorW is anchored to the OM, is localized in the periplasm, and appears to be monomeric.** Bioinformatic analyses using the SignalP prediction algorithms identified that *P. gingivalis* PorW possesses a type II signal peptide (SPII) (Fig. 4A), typical of lipoproteins. Indeed, all members of the PorW family share a typical lipobox sequence with an N-terminal cysteine residue (position +1) that is acylated to anchor the processed protein to the membrane (Fig. 4B). Rhodes et al. have shown that SprE (PorW

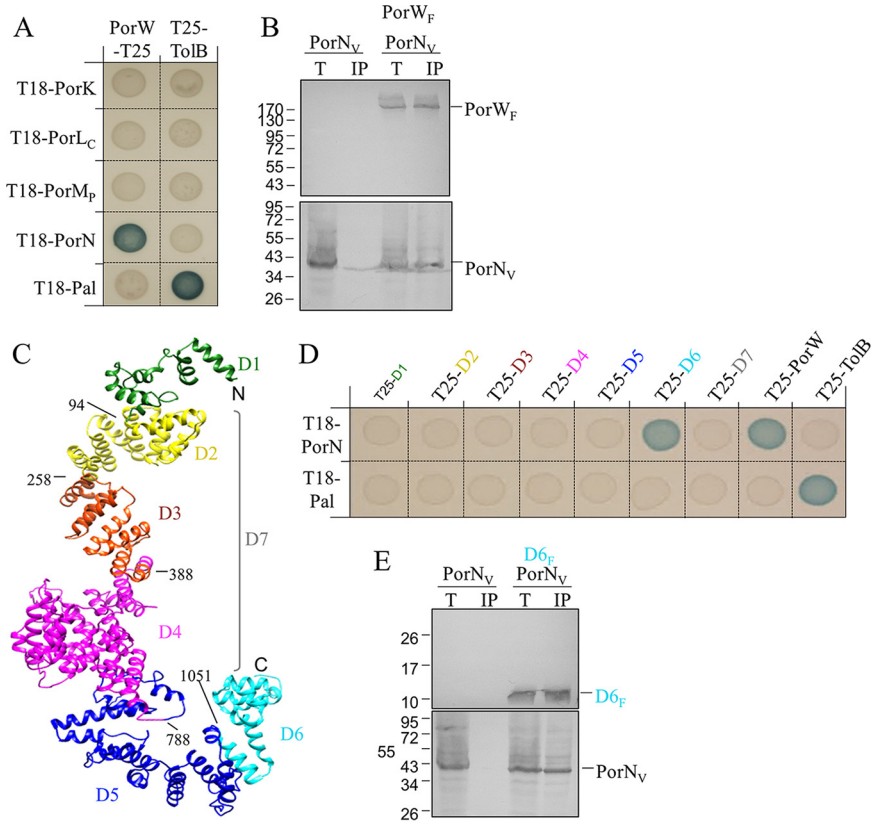

**FIG 3** PorN interacts with the PorW C-terminal domain (D6). (A) BACTH assay. BTH101 cells producing the indicated proteins or domains (PorL$_C$, cytoplasmic domain of PorL, amino acids 73 to 309; PorM$_P$, periplasmic domain of PorM, amino acids 36 to 516) fused to the T18 and T25 domain of the *Bordetella* adenylate cyclase were spotted on X-Gal-IPTG reporter LB agar plates. The blue color of the colony reports interaction between the two partners. Controls include T18 and T25 fusions to TolB and Pal, two proteins that interact but are unrelated to the T9SS. (B) Coimmunoprecipitation assays. Detergent-solubilized lysates of *E. coli* cells producing PorW$_F$ and VSV-G-tagged PorN (PorN$_V$, panel B) were subjected to immunoprecipitation with anti-FLAG-coupled beads. The total lysates (T) and immunoprecipitated (IP) material were separated by 10% SDS-PAGE and immunodetected with anti-FLAG (upper panels) and anti-VSV-G (lower panels) monoclonal antibodies. Molecular weight markers are indicated on the left. (C) Domain segmentation of the *P. gingivalis* PorW structural model. Each domain is shown in a different color (D1, green; D2, yellow; D3, red; D4, pink; D5, blue; D6, cyan). The D7 region (indicated by the gray bracket) corresponds to domains D2-D3-D4. The boundaries of the domains are indicated. (D) BACTH assay. BTH101 cells producing the indicated proteins or domains fused to the T18 and T25 domain of the *Bordetella* adenylate cyclase were spotted on X-Gal-IPTG reporter LB agar plates. The blue color of the colony reports interaction between the two partners. Controls include T18 and T25 fusions to TolB and Pal, two proteins that interact but are unrelated to the T9SS. (E) Coimmunoprecipitation assay. Soluble lysates of *E. coli* cells producing FLAG-tagged D6 domain of PorW (D6$_F$) and VSV-G-tagged PorN (PorN$_V$) were subjected to immunoprecipitation with anti-FLAG-coupled beads. The total lysates (T) and immunoprecipitated (IP) material were separated by 13% SDS-PAGE and immunodetected with anti-FLAG (upper panel) and anti-VSV-G (lower panel) monoclonal antibodies. Molecular weight markers are indicated on the left.

homologue in *F. johnsoniae*) is located in the membrane (37), and with the presence of a serine residue at the +2 position (Fig. 4B), these analyses also suggest that the PorW final destination is the OM, as Ser or Gly residues are generally located adjacent to the +1 attachment cysteine in *Bacteroidetes* OM lipoproteins (38). To gain further information on PorW localization and to distinguish whether PorW is OM-embedded or a peripheral OM protein attached to either the outer side or the periplasmic side of the OM, we used a proteinase K susceptibility assay. Whole cells and lysed cells were treated with proteinase K, and aliquots taken at various time points were subjected to immunoblot analysis. Anti-PorW and anti-PorK reactive bands were present in the fractions that were not lysed for up to 4 h (Fig. 4C). However, in the lysed fractions, PorW and PorK bands were observed only at the zero time point (Fig. 4C). GroEL is a

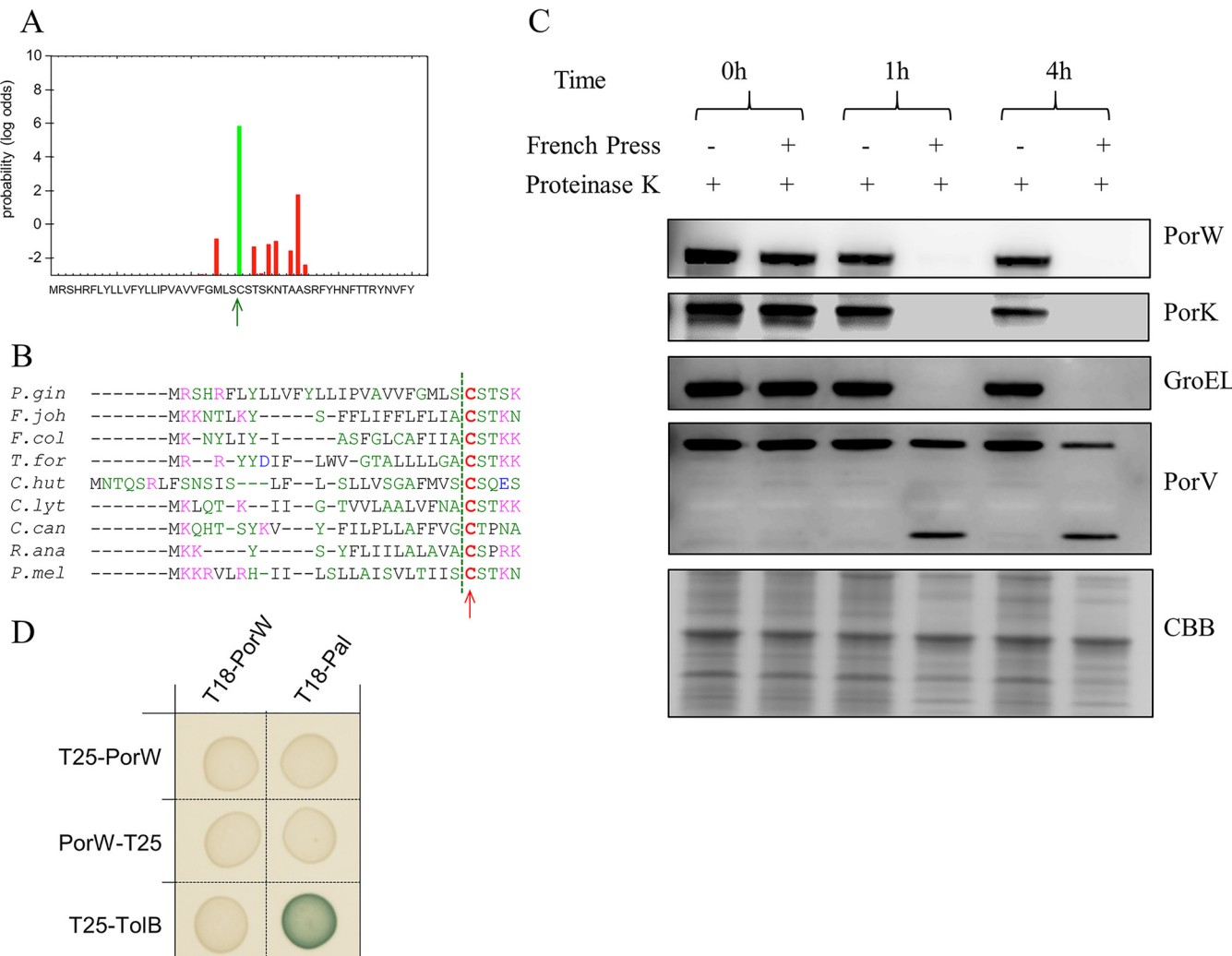

**FIG 4** The PorW subunit is likely to be a monomeric OM lipoprotein exposed in the periplasm. (A) PorW has a predicted type II signal sequence. Graph reporting the probability of cleavage at the indicated position within the N-terminal sequence by signal peptidase I (red bars) and signal peptidase II (green bar) using the SignalP and LipoP programs. The most probable cleavage site (between residues Ser and Cys) is indicated by the green arrow. (B) Sequence alignment of N-terminal sequences of selected PorW/SprE homologues from strains indicated on left (*P.gin*, *Porphyromonas gingivalis*; *F.joh*, *Flavobacterium johnsoniae*; *F.col.*, *F. columnare*; *T.for*, *Tannerella forsythia*; *C.hut*, *Cytophaga hutchinsonii*; *C.lyt*, *Cellulophaga lytica*; *C.can*, *Capnocytophaga canimorsus*; *R.ana*, *Riemerella anatipestifer*, *P.mel*, *Prevotella melaninogenica*) highlighting the position of the predicted type II signal peptidase cleavage sites (dashed green vertical line, SignalP and LipoP programs) and the putative Cys +1 residues (red letters) of the mature forms. (C) *P. gingivalis* ABK⁻ cells were either lysed using a French Pressure cell or left unlysed. Cells were treated with proteinase K, and aliquots were collected at 0, 1, and 4 h. Samples were subjected to SDS-PAGE followed by immunoblot analysis using antibodies against the protein indicated on the right. The Coomassie blue-stained gel (CBB) shows the relative loading amount. (D) BACTH assay. BTH101 cells producing the indicated proteins fused to the T18 and T25 domain of the *Bordetella* adenylate cyclase were spotted on X-Gal-IPTG reporter LB agar plates. The blue color of the colony reports interaction between the two partners. Controls include T18 and T25 fusions to TolB and Pal, two proteins that interact but are unrelated to the T9SS.

cytoplasmic protein, and the immunoblot profile was similar to that of PorW (Fig. 4C). PorV is an integral OM protein (39), and anti-PorV reactive bands appeared in all fractions (Fig. 4C). However, in the lysed fractions at 1 and 4 h, a PorV band at a lower molecular weight was also observed (Fig. 4C), suggesting that perhaps one of the periplasmic loops of PorV is susceptible to proteinase K. The extreme sensitivity of PorW to proteinase K after cell lysis together with its resistance to cleavage in whole cells is consistent with a location on the periplasmic side of the OM. Next, we tested whether PorW oligomerizes using BACTH. PorW did not interact with itself under the BACTH conditions, suggesting that the PorW subunit is likely to be monomeric (Fig. 4D).

**PorD interacts with domain 3 of PorW and is required for the efficient function of the T9SS.** Our previous analyses suggested that PorD associates with Sov and PorW in a PorW-dependent manner. To test the interaction between PorD and PorW, we

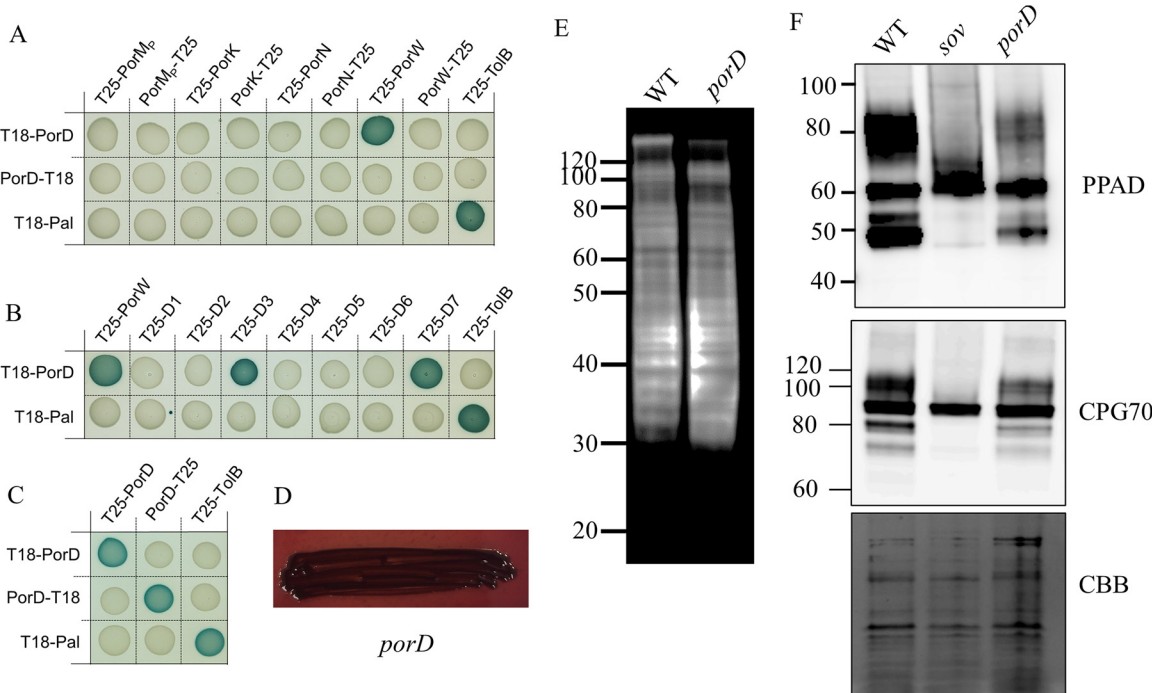

**FIG 5** PorD interacts with domain 3 of PorW and is required for full efficiency of the T9SS. (A to C) BACTH assay of PorD with components of the PorKLMN trans-envelope complex and PorW (A), with separate PorW domains (B), and with PorD (C). BTH101 cells producing the indicated proteins fused to the T18 and T25 domain of the *Bordetella* adenylate cyclase were spotted on X-Gal-IPTG reporter LB agar plates. The blue color of the colony reports interaction between the two partners. Controls include T18 and T25 fusions to TolB and Pal, two proteins that interact but are unrelated to the T9SS. (D) Black pigmentation of *porD* deletion mutant on blood agar plate. Proteins in the whole-cell lysates of WT and *porD* deletion mutant were electrophoresed on an SDS-PAGE gel. (E and F) The proteins were transferred onto a membrane and probed with MAb-1B5 (anti-A-LPS) (E), as well as PPAD and CPG70 antibodies (F). The Coomassie blue-stained gel (CBB) shows the relative loading amount.

performed bacterial two-hybrid experiments. Additionally, the interactions of PorD with PorK, PorM, and PorN were also tested. PorD is a predicted periplasmic protein (40) and was cloned into the BACTH vectors without its signal peptide. The BACTH assays showed that PorD interacts with PorW but not PorK, PorM, or PorN (Fig. 5A). Next, we assessed which region of the PorW protein associated with PorD. Blue colonies were observed with PorW domain 3 (aa 259 to 388) and domain 7 (aa 95 to 788) (Fig. 5B). Together, this indicates that PorD interacts with domain 3 of PorW. We also tested if PorD oligomerized using BACTH assays. PorD interacted with itself (Fig. 5C), suggesting that PorD is likely to associate with PorW in a multimeric state. To examine if PorD is required for the function of the T9SS, we created a *porD* deletion mutant in *P. gingivalis*. The *porD* mutant was black pigmented (Fig. 5D), and an immunoblot with A-LPS antibodies showed high-molecular-weight A-LPS in the *porD* mutant, consistent with the presence of secreted and attached cargo proteins (Fig. 5E). To determine whether there may be a slowing down of secretion in the *porD* mutant, we performed immunoblots on whole-cell lysates from *P. gingivalis* wild-type (WT), *sov* and *porD* mutant cells and probed with antibodies specific to the cargo proteins, PPAD and CPG70. This type of analysis is particularly suited because the precursor levels of PPAD (~60 kDa) and CPG70 (~90 kDa) were similar in the WT, *sov* and *porD* mutant cells; however, the modified PPAD (~70 to 90 kDa) and CPG70 (100 to 120 kDa) were abundant in WT cells, absent in *sov* mutant cells, and at intermediate levels in *porD* mutant cells (Fig. 5F). Furthermore, the N and C termini-processed PPAD (~47 kDa) and CPG70 (~80 kDa) (41) in the WT were increased dramatically compared to those in the *porD* mutant (Fig. 5F). Collectively, the number of these cargo proteins being secreted and processed is reduced in the *porD* mutant relative to that in the WT, indicating that PorD is required for the efficient operation of the T9SS.

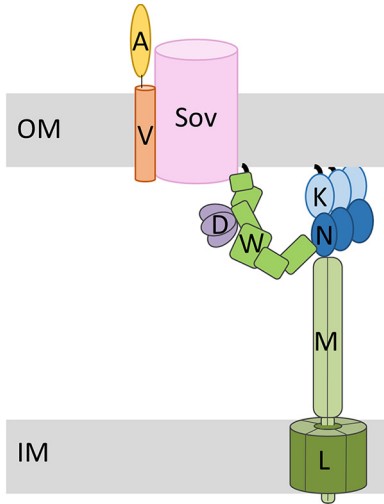

**FIG 6** Proposed model of T9SS organization. The Sov translocon in the OM interacts directly with PorV bound to the cargo protein PorA. The PorL/M motor complex in the IM connects to the PorK/N ring tethered to the OM. The PorW lipoprotein connects the Sov translocon to the PorK/N ring complex. The PorD multimer interacts with PorW. outer membrane (OM), inner membrane (IM), PorW (W), PorD (D), PorK (K), PorN (N), PorL (L), PorM (M), PorV (V), and PorA (A).

## DISCUSSION

The T9SS of *P. gingivalis* is a secretion workhorse translocating and attaching over 30 cargo proteins to the cell surface, where they form an electron-dense surface layer surrounding the whole cell (39). To do this, the T9SS requires an energy source and a means of efficiently channeling the cargo to the secretion pore. The T9SS is composed of subcomplexes with specific functions: the PorKLMN trans-envelope complex comprising the PorL/M motor that energizes cargo secretion and the PorK/N ring, the Sov translocon through which cargo are secreted, and the attachment complex that anchors cargo at the cell surface. This study demonstrates that the PorW OM lipoprotein bridges the translocon to the PorK/N ring and suggests a pathway by which energy and substrates may be directed all the way from the inner membrane motor complex to the secretion pore (Fig. 6). This finding was made through studying the protein interactome of the Sov translocon in *P. gingivalis* and testing pairwise interactions in the heterologous host *E. coli*. In addition, novel interactions were also observed, including a PorA-PorV-Sov complex and the identification of a new T9SS component, PorD which binds to PorW.

The structures of the SprA translocon bound to either PorV or the Plug in *F. johnsoniae* were recently solved (25). Peptidyl-prolyl isomerase (PPI) was found to associate with SprA in both PorV and Plug bound states. In our study, PPI was not identified to interact with Sov; however, surprisingly, we found PorA to be part of the PorV-Sov complex in *P. gingivalis*. PorA was recently shown to be an essential component of the T9SS and shown to be involved in signal transduction as part of the PorXY-SigP regulatory system (26). PorA is also a cargo protein that, like other cargo, is secreted, binds to PorV, has its CTD signal removed, and is conjugated to A-LPS (31, 32). Our finding that PorA is a core component of the most abundant Sov complex, PorA-PorV-Sov, suggests that PorA may play its essential T9SS role in this form. We have drawn PorA on the cell surface (Fig. 6), because as a cargo protein it is expected to be secreted through the Sov channel and bind to the outside surface of PorV. However, while most cargo at this point are thought to be shuttled to the attachment complex, it appears that some PorA molecules remain bound to PorV-Sov. The PorA-PorV-Sov complex was also observed in T9SS mutants (Table 1), and therefore the surface location of PorA in these strains is uncertain. Strangely, there could be an alternative secretion pathway for PorA since it was apparently exposed at the cell surface in both the *sov* and *porV* mutants (26).

PGN_1783 (PorD) has twice been predicted to be a component of the T9SS (3, 20) but was previously discounted as such because its deletion in *P. gingivalis* did not result in loss of black-pigmented colonies (3). Here, PorD was concluded to be a genuine T9SS component since it binds to PorW and its deletion reduced the efficiency of secretion. Furthermore, the Sov interactome was altered in the *porD* mutant with a high level of Plug detected and the lowest level of PorV and PorA detected (Table 1). A possible explanation is that the dissociation of Plug from the Sov translocon may be induced by the binding of PorW/PorD to Sov. Without PorD, Plug removal would be slower; hence, with less of Sov being in the Plug-free form, secretion would be slower. However, in T9SS mutants where secretion is completely blocked, it appears that the translocon is mostly in the PorV-bound form (PorA-PorV-Sov) (Table 1).

The major finding of this study concerns the PorW lipoprotein. The presence of a specific SPII cleavage site followed by a conserved cysteine residue suggests that the N terminus of the mature form of PorW is lipidated and embedded into the OM. After PorK and PorE, PorW is the third T9SS OM lipoprotein to be characterized. The major role of lipoproteins in the T9SS explains why LolA, a protein of the Lol pathway involved in lipoprotein sorting, is important for motility in *F. johnsoniae* (37). Having three independent, conserved, and essential lipoproteins in a secretion system is uncommon but could be explained in the case of the T9SS by the multiple subcomplexes that need to be stably anchored to the OM.

The homologue of PorW in *F. johnsoniae* is SprE which was localized to the OM (37). PorW was accessible to proteinase K cleavage only after cell lysis (Fig. 4C), consistent with a location underneath the OM, with periplasmic exposure. Such a position is in agreement with the results that PorW interacts with two periplasmic proteins, PorN and PorD (Fig. 6). Our domain analysis coupled to protein-protein interaction assays showed that the C-terminal D6 domain of PorW is required and sufficient for binding to PorN. PorD was shown to bind to the PorW D3 domain. Although not yet determined, we postulate that Sov might bind to the most N-terminal domains, D1 or D2. The cryo-electron microscopy (cryo-EM) structure of SprA/Sov indicated it to be a monomer (25), and our BACTH assay did not detect any PorW oligomerization, suggesting that the Sov-PorW interaction is 1:1.

Interestingly, the PorW D6 domain is not conserved in PorW/SprE proteins: while present in the PorW homologues of *Tannerella* and *Prevotella*, this C-terminal extension is absent in *F. johnsoniae* SprE and SprE/PorW homologues in *Flavobacteria*, *Cytophaga*, and *Cellulophaga*. The PorKN-tethering function of PorW is therefore unlikely to be conserved in all T9SS nanomachines. A number of additional lipoproteins associated with the gliding machinery in *Bacteroidetes*, such as GldB, GldD, GldH, GldI, or GldJ (6), may compensate for the absence of the D6 domain.

It is interesting to speculate on the function of PorW in tethering the PorK/N ring to the Sov translocon. First, we propose that PorW might be important to guide the cargo to the secretion pore. We can also suggest that PorW might be important for energy transduction from the PorL/M motor to the translocon. It is now established that the PorL/M complex uses the proton-motive force to power cargo secretion through the T9SS. The PorM rotor component of the PorL/M motor complex has a long, rotating periplasmic arm that is predicted to drive secretion (and also gliding motility in other species) via the PorK and PorN ring proteins (30). However, it is not determined whether the mechanical energy derived from the PMF is transduced to the PorK/N ring or further. Previously, it was shown that PorM interacts with PorN (28), and in this study, we show that PorN also interacts with PorW and PorW interacts with the Sov translocon. We therefore propose that PorM-PorN-PorW form the powertrain transducing energy derived from the proton-motive force at the inner membrane to the secretion pore (Fig. 6). Since each T9SS comprises a large PorK/N ring comprising up to 36 subunits each but appears to have only one Sov and one PorW (this study, reference 42), the T9SS as a whole may be asymmetric where the Sov translocon is not in the center of the ring but to one side, with the ring more like a slide carousel delivering

slides to the projector window. One PorW bridge could place Sov either inside the ring or outside the ring.

The way in which energy is transduced through the ring requires further investigation. We speculate that the energy may be transferred through PorN rotation on a PorK track and at one point in the circle, a PorN subunit is driven against PorW, causing PorW to adopt a high-energy conformation. The energy is then transferred through the PorW molecule to Sov to energize secretion (Fig. 6). Another interesting point is that both PorM and PorN have been shown to bind to the CTD signal of cargo proteins (43). The rotating arms of PorM could potentially collect cargo proteins from the periplasm and transfer them to PorN. PorN may then transfer them to PorW and Sov. The energetic process of concentrating substrates at the entrance of the secretion pore may be an alternative or additional means of powering secretion.

In conclusion, we demonstrate for the first time that the T9SS PorK/N ring is linked to the Sov translocon via PorW, suggesting that this interaction may be the pathway for the transduction of energy and delivery of cargo to the secretion pore (Fig. 6).

## MATERIALS AND METHODS

**Bacterial strains and culture conditions.** Strains used in this study are listed in Table S1. *P. gingivalis* wild-type strain ATCC 33277 was grown in tryptic soy-enriched brain heart infusion broth (TSBHI) (25 g/L tryptic soy, 30 g/L BHI) supplemented with 0.5 mg/mL cysteine, 5 $\mu$g/mL hemin, and 5 $\mu$g/mL menadione. For blood agar plates, 5% defibrinated horse blood (Equicell, Bayles, Australia) was added to enriched Trypticase soy agar. Mutant strains were grown in the same media as listed above with the appropriate antibiotic selections. All *P. gingivalis* strains were grown anaerobically (80% $N_2$, 10% $H_2$, and 10% $CO_2$) at 37°C. *Escherichia coli* DH5$\alpha$ (New England Biolabs), W3110 (EC laboratory collection), and BTH101 (44) were used for cloning procedures, coimmunoprecipitation, and bacterial two-hybrid assays, respectively. *E. coli* cells were routinely grown aerobically in lysogeny broth (LB) at 30°C or 37°C. Plasmids were maintained by addition of ampicillin (100 $\mu$g/mL) or kanamycin (50 $\mu$g/mL). When necessary, gene expression was induced with 0.1 to 0.5 mM isopropyl-$\beta$-thio-galactoside (IPTG), 2 mg/mL of L-arabinose, or 0.05 $\mu$g/mL of anhydrotetracycline (AHT). Strains used in this study are listed in Table S1.

**Plasmid construction.** Plasmids used for this study are listed in Table S1. BACTH, pASK-IBA, and pBAD vectors encoding the PorKLMN proteins or domains have been described previously (28, 29, 45). BACTH, pASK-IBA, and pBAD vectors encoding full-length PorW, processed PorW, PorW domains, Sov, or PorD were constructed by standard restriction/ligation cloning or restriction-free cloning (46) as described previously (47). DNA fragments were amplified from *P. gingivalis* genomic DNA extracted from 6 × 10⁹ cells using a DNA purification kit (DNeasy Blood & Tissue, Qiagen). PCRs were performed with a Biometra thermocycler using Q5 DNA polymerase (New England Biolabs) and custom oligonucleotides synthesized by Sigma-Aldrich (listed in Table S1). Restriction/ligation cloning was performed by digesting DNA fragments with restriction enzymes (New England Biolabs) and then ligating them using T4 DNA ligase (New England Biolabs) into target vectors that had been digested with the same enzymes and dephosphorylated (Shrimp Alkaline Phosphatase, New England Biolabs). For restriction-free cloning, DNA fragments were amplified using primers that introduced extensions capable of annealing to the target vector. The double-stranded product of the first PCR was then used as primer for a second PCR using the target vector as the template. PCR products were then treated with DpnI to eliminate template plasmids and transformed into DH5$\alpha$-competent cells. All the constructs were verified by colony PCR and DNA sequencing (Eurofins sequencing).

**Deletion of *pgn_1783* from *P. gingivalis* ATCC 33277.** As *pgn_*1783 (*porD*) is the fourth gene of a predicted seven gene operon, the open reading frame (ORF) of the erythromycin resistance gene *ermF* (48) was used to replace the *porD* ORF. A recombination cassette was designed where 500 bp from each side of the *porD* ORF was used to flank the *ermF* ORF. The recombination cassette was manufactured by Integrated DNA Technologies (IDT) and supplied in their pUCIDTAmp vector. This plasmid was linearized with AatII prior to electroporation into *P. gingivalis* ATCC 33277. Transformants were selected on blood agar plates containing erythromycin. Primers *pgn_1784* F (5'-CATACGAGAAACTCGATCCTG-3') and *pgn_*1782 R (5'-TGCCAATAGTCATACTCCGAC-3') were designed to bind *P. gingivalis* ATCC 33277 genomic DNA outside the recombination cassette and were used to confirm replacement of *porD* with *ermF*.

**Antibodies.** The work below was done by Genscript according to their SC1676 package. The N-terminal (33 to 240) and C-terminal (2,281 to 2,499) amino acid regions of *P. gingivalis* Sov, the N-terminal (28 to 398) amino acid regions of *P. gingivalis* PorW, and the amino acid region of 123 to 456 of *P. gingivalis* PPAD were expressed in *E. coli* and purified. Rabbit polyclonal antisera were raised against these antigens and the antigens were used to affinity purify the antibodies.

**Blue native gel electrophoresis.** BN-PAGE was performed essentially as described previously (3). Briefly, *P. gingivalis* cells were pelleted by centrifugation at 5,000 × *g* for 5 min at 4°C and the pellet was suspended in native gel buffer containing 1% *n*-dodecyl-$\beta$-D-maltoside (DDM), complete protease inhibitors, and 5 mM MgCl₂. After sonication (3), the samples were clarified by centrifugation at 16,000 × *g* for 20 min at 4°C. Coomassie blue G-250 was added to the samples at a final concentration of 0.25%, and the samples were electrophoresed on nondenaturing Native PAGE Novex 3 to 12% Bis-Tris gels. The proteins in the gels were either transferred onto a polyvinylidene difluoride (PVDF) membrane for

immunodetection with antibodies specific for Sov as per the immunoblot method below (3) or stained with Coomassie G-250 and destained until the background was clear. For the Coomassie-stained gel, the lane containing the sample was excised into 12 gel bands and in-gel digestion was performed. The gel pieces were incubated with 2% SDS and 10 mM dithiothreitol (DTT) at 56°C for 1 h. Following incubation, 55 mM iodoacetamide was added to the gel pieces and incubated for 30 min in the dark. In-gel digestion was performed using sequencing-grade-modified trypsin (Promega) and incubated overnight at 37°C, as published previously (49). Tryptic peptides were extracted from the gel pieces using 50% acetonitrile in 0.1% trifluoroacetic acid (TFA) and sonicated for 10 min in a sonicator bath. The samples were concentrated in a vacuum centrifuge before analysis using liquid chromatography-tandem mass spectrometry (LC-MS/MS).

**LC-MS/MS.** The tryptic peptides were analyzed by LC-MS/MS using the Q Exactive Plus Orbitrap mass spectrometer coupled to an ultimate 3000 UHPLC system (Thermo Fisher Scientific). Buffer A was 2% acetonitrile and 0.1% formic acid, and buffer B consisted of 0.1% formic acid in acetonitrile. Sample volumes of 1 $\mu$L were loaded onto a PepMap $C_{18}$ trap column (75 $\mu$m internal diameter [ID], 2 cm, 100 Å) and desalted at a flow rate of 2 $\mu$L/min for 15 min using buffer A. The samples were then separated through a PepMap $C_{18}$ analytical column (75 $\mu$m ID, 15 cm, 100 Å) at a flow rate of 300 nL/min, with the percentage of solvent B in the mobile phase changing from 2% to 10% in 1 min, from 10% to 35% in 50 min, from 35% to 60% in 1 min, and from 60% to 90% in 1 min. The spray voltage was set at 1.8 kV, and the temperature of the ion transfer tube was 250°C. The S-lens was set at 50%. The full MS scans were acquired over an $m/z$ range of 300 to 2,000, with a resolving power of 70,000, an automatic gain control (AGC) target value of $3 \times 10^6$, and a maximum injection of 30 ms. Dynamic exclusion was set at 90 s. Higher-energy collisional dissociation MS/MS scans were acquired at a resolving power of 17,500, AGC target value of $5 \times 10^4$, maximum injection time of 120 ms, isolation window of $m/z$ 1.4, and NCE of 25% for the top 15 most abundant ions in the MS spectra. All spectra were recorded in profile mode.

Relative abundances of proteins were quantified by MaxQuant (version 1.5.3.30) (50). Raw MS/MS files were searched against the *P. gingivalis* ATCC 33277 database. The default MaxQuant parameters were used, except LFQ minimum ratio count was set to 1 and the match between runs was selected. MaxQuant normalized the data set as part of data processing. For analysis of the BN-PAGE profiles, LFQ intensities were used. For quantification of the immunoprecipitated material, the LFQ intensity ratio test strain/control strain was used to identify proteins that were significantly enriched. To determine the relative abundance of the enriched proteins, the corrected iBAQ intensity (test strain-control strain) was calculated and expressed as a percentage relative to the target protein.

**Bacterial two-hybrid assays.** *Bordetella* adenylate cyclase-based bacterial two-hybrid assays (44) were performed as described previously (28). Briefly, the proteins to be tested were fused to the isolated T18 and T25 catalytic domains of the *Bordetella* adenylate cyclase. After introduction of the two plasmids producing the fusion proteins into the reporter BTH101 strain, plates were incubated at 30°C for 24 h. Three independent colonies for each transformation were inoculated into 600 $\mu$L of LB medium supplemented with ampicillin, kanamycin, and 0.5 mM IPTG. After overnight growth at 30°C, 10 $\mu$L of each culture was spotted onto LB plates supplemented with ampicillin, kanamycin, IPTG, and 40 $\mu$g/mL of 5-bromo-4-chloro-3-indolyl $\beta$-D-galactopyranoside (X-gal, Euromedex) and incubated for 6 to 16 h at 30°C. Controls included interaction assays with TolB/Pal, a protein pair unrelated to the T9SS. The experiments were done at least in triplicate, and a representative result is shown.

**Proteinase K susceptibility assay.** Proteinase K susceptibility assay was performed per previously published protocol (27).

**Immunoblots.** Cell lysates from *P. gingivalis* WT or mutants were separated by SDS-PAGE, and the proteins were transferred onto a nitrocellulose membrane. The membrane was blocked in 5% skim milk and probed with MAb-1B5 (a kind gift from M.A. Curtis) (51), Sov-, PPAD- or CPG70-specific antibodies, or commercial anti-FLAG (clone M2, Sigma-Aldrich), and anti-VSV-G (clone P5D4, Sigma-Aldrich) monoclonal antibodies followed by anti-mouse or anti-rabbit HRP (or alkaline phosphatase)-conjugated secondary antibodies. HRP was developed using SuperSignal West Pico chemiluminescent substrate and visualized with an LAS-3000 imaging system. Alkaline phosphatase signal was developed in alkaline buffer in the presence of 5-bromo-4-chloro-3-indolylphosphate and nitroblue tetrazolium.

**Coimmunoprecipitation in *P. gingivalis*.** *P. gingivalis* cells were lysed in 20 mM Tris-HCl (pH 8), 100 mM NaCl, 1% DDM, tosyl-L-lysyl-chloromethane hydrochloride (TLCK), and complete protease inhibitors. The cells were then sonicated as per BN-PAGE protocol and clarified by centrifugation at 16,000 × $g$ for 10 min at 4°C. Cell lysates were mixed with Sov antibodies bound to agarose. The beads were rotated on a wheel for 2 h at 4°C and then washed with 20 mM Tris-HCl (pH 8), 100 mM NaCl, followed by 300 mM high salt washes, and lastly with 10 mM Tris-HCl (pH 8). The beads were suspended in Laemmli buffer and boiled for 10 min. The samples were subjected to SDS-PAGE electrophoresis for a short period of time and stained with Coomassie blue. The sample was excised as a single band and subjected to tryptic digestion and LC-MS/MS analysis.

**Coimmunoprecipitation in the heterologous host *E. coli*.** Coimmunoprecipitation in *E. coli* was performed as described previously (52) with modifications. W3110 cells producing the proteins of interest were grown to an absorbance at 600 nm ($A_{600}$) of ~0.4, and the expression of the cloned genes was induced with L-arabinose or AHT for 0.75 to 1 h. Then, $1.5 \times 10^{10}$ cells were harvested, and the pellets were resuspended in 1 mL of CelLyticB buffer (Sigma-Aldrich) supplemented with 100 $\mu$g/mL lysozyme, 100 $\mu$g/mL DNase, and protease inhibitors (Complete, Roche). After incubation for 30 min at 25°C, lysates were clarified by centrifugation at 10,000 × $g$ for 10 min. For PorW-D6/PorN interactions, cell lysates were treated with 1% *para*-formaldehyde for 45 min to stabilize the interactions. The cell lysates

were mixed with anti-FLAG M2 affinity gel (Sigma-Aldrich). After 1 h of incubation, the beads were washed three times with 1 mL of 20 mM Tris-HCl (pH 7.5), 100 mM NaCl, and 0.5% Triton X-100 and once with 1 mL of 20 mM Tris-HCl (pH 7.5), 100 mM NaCl, and 0.05% Triton X-100. Beads were resuspended in 25 $\mu$L of Laemmli loading buffer, boiled for 10 min, and subjected to SDS-PAGE and immunodetection analyses.

**Bioinformatics analyses.** The SignalP 5.0 (53), LipoP 1.0 (54), Phyre2 (55), Pfam (56), and TPRPred (57) computer algorithms were used to predict signal peptides, lipoboxes, tertiary structure, domain architecture, and TPR motifs, respectively.

## SUPPLEMENTAL MATERIAL

Supplemental material is available online only.

**SUPPLEMENTAL FILE 1**, PDF file, 0.1 MB.

## ACKNOWLEDGMENTS

We thank the members of the Reynolds and Cascales laboratories for discussions. We thank the Research Transfer Facility at Bio21 Molecular Science and Biotechnology Institute, The University of Melbourne, for mass spectrometer facilities.

The work in E.C.R.'s laboratory was supported by the Australian National Health and Medical research Council grants ID 1193647 and ID 1123866, the Australian Research Council grant DP200100914, and the Australian Dental Research Foundation grant 349-2018. The work on T9SS in E.C.R.'s laboratory was funded by the Centre National de la Recherche Scientifique, the Aix-Marseille Université, and grants from the Agence Nationale de la Recherche (ANR-15-CE11-0019, ANR-20-CE11-0011) and from the Excellence Initiative of Aix-Marseille Université, a French "Investissements d'Avenir" program (A*MIDEX AAP-ID-17-33). The PhD thesis of I.L.S. has been funded by the ANR grant.

D.G.G.: conceptualization, methodology, formal analysis, data curation, writing - original draft. C.A.B.: methodology. I.L.S.: methodology, formal analysis, data curation. M.C. and T.D.: methodology, supervision. E.C., P.D.V., E.C.R.: conceptualization, supervision, funding acquisition, writing - review and editing.

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
