## [Reviewer comments · Microbiology Spectrum]

Microbiology Spectrum

Protein interactome analysis of the type IX secretion system identifies PorW as the missing link between the PorK/N ring complex and the Sov translocon

Dhana Gorasia, Ignacio Lunar Silva, Catherine Butler, Maïalène Chabalier, Thierry Doan, Eric Cascales, Paul Veith, and Eric Reynolds

Corresponding Author(s): Eric Reynolds, University of Melbourne

Review Timeline:

Submission Date:	September 16, 2021
Editorial Decision:	November 18, 2021
Revision Received:	November 25, 2021
Accepted:	December 7, 2021

Editor: Cezar Khursigara

Reviewer(s): The reviewers have opted to remain anonymous.

Transaction Report:

DOI: <https://doi.org/10.1128/spectrum.01602-21>

November 18, 2021

Prof. Eric C Reynolds
University of Melbourne
Oral Health CRC, Melbourne Dental School, Bio21 Institute
720 Swanston Street
Melbourne, Victoria 3010
Australia

Re: Spectrum01602-21 (Protein interactome analysis of the type IX secretion system identifies PorW as the missing link between the PorK/N ring complex and the Sov translocon)

Dear Prof. Eric C Reynolds:

Two experts in the field and I have now reviewed your manuscript and we are all in agreement that this is a well written paper with novel findings. Reviewer #2 has suggested additional experiments and although I agree that these would be nice to include, they are not required for acceptance of the paper. Please focus on the changes suggested to the text.

Thank you for submitting your manuscript to Microbiology Spectrum. As you will see your paper is very close to acceptance. Please modify the manuscript along the lines I have recommended. As these revisions are quite minor, I expect that you should be able to turn in the revised paper in less than 30 days, if not sooner. If your manuscript was reviewed, you will find the reviewers' comments below.

When submitting the revised version of your paper, please provide (1) point-by-point responses to the issues I raised in your cover letter, and (2) a PDF file that indicates the changes from the original submission (by highlighting or underlining the changes) as file type "Marked Up Manuscript - For Review Only". Please use this link to submit your revised manuscript. Detailed instructions on submitting your revised paper are below.

Link Not Available

Sincerely,

Cezar Khursigara

Reviewer comments:

Reviewer #1 (Comments for the Author):

This is a very clearly written manuscript that follows through on previous work by the laboratory. This group's experience in studying the type IX secretion system greatly facilitated the identification of PorW as a key bridging component connecting the inner and outer membrane. The work was clearly presented with appropriate controls. For example, providing the rationale for the choice of gingipain mutants to conduct the immunoblot analyses. The work is significant in that understanding this key secretion apparatus of *P. gingivalis* may facilitate highly specific novel inhibitors that may be of clinical value.

Reviewer #2 (Comments for the Author):

This is an exceptionally well written manuscript summarizing novel and interesting findings regarding the interactome of T9SS components. The identification of PorW as a missing link between the PorL/M motor and the Sov translocon casts a new light on energy transduction from the inner membrane to the secretion pore.

The points needed to be addressed:

1. PorD the now confirmed T9SS component is encoded by the fourth gene (pgn_1783) in a predicted seven gene operon. In the Δ porD mutant the gene was replaced with ermF cassette. It is not clear how this genetic manipulation affected expression of downstream genes. This is important in the light of the authors' conclusion that PorD although non-essential for T9SS functioning contributes to efficiency of cargo protein secretion.
2. Methods are described in details therefore salt concentration used to wash beads charged with Sov antibodies and used for immunoprecipitation should be provided (lane 223).
3. The results indicating PorW is required for Sov to form a high molecular weight complex are not convincing (Page 12) and should be verified by western blot analysis using anti-PorW antibodies. Alternatively cross-linking followed up SDS-PAGE and MS analysis should be considered.
4. Importance of PorD for the efficient operation of the T9SS needs to be verified qualitatively by comparing the activity of gingipains and PPAD, alternatively CPG70 secreted by WT-Pg and Δ porD.

Preparing Revision Guidelines

- point-by-point responses to the issues I raised in your cover letter
- Upload a compare copy of the manuscript (without figures) as a "Marked-Up Manuscript" file.
- Each figure must be uploaded as a separate file, and any multipanel figures must be assembled into one file.
- Manuscript: A .DOC version of the revised manuscript
- Figures: Editable, high-resolution, individual figure files are required at revision, TIFF or EPS files are preferred

Please return the manuscript within 60 days; if you cannot complete the modification within this time period, please contact me. If you do not wish to modify the manuscript and prefer to submit it to another journal, please notify me of your decision immediately so that the manuscript may be formally withdrawn from consideration by Microbiology Spectrum.

Reviewer #1 (Comments for the Author):

This is a very clearly written manuscript that follows through on previous work by the laboratory. This groups experience in studying the type IX secretion system greatly facilitated the identification of PorW as a key bridging component connecting the inner and outer membrane. The work was clearly presented with appropriate controls. For example, providing the rationale for the choice of gingipain mutants to conduct the immunoblot analyses. The work is significant in that understanding this key secretion apparatus of *P. gingivalis* may facilitate highly specific novel inhibitors that maybe of clinical value.

We greatly appreciate the reviewers time in reading our manuscript.

Reviewer #2 (Comments for the Author):

This is exceptionally well written manuscript summarizing novel and interesting findings regarding the interactome of T9SS components. The identification of PorW as a missing link between the PorL/M motor and the Sov translocon casts a new light on energy transduction from the inner membrane to the secretion pore.

The points needed to be addressed:

1. PorD the now confirmed T9SS component is encoded by the fourth gene (pgn_1783) in a predicted seven gene operon. In the Δ porD mutant the gene was replaced with ermF cassette. It is not clear how this genetic manipulation affected expression of downstream genes. This is important in the light of the authors' conclusion that PorD although non-essential for T9SS functioning contributes to efficiency of cargo protein secretion.

The plasmid was designed carefully to ensure none of the other genes in the operon were affected. The correct insertion of the ermF cassette was verified by PCR. Additionally, PorD is the only gene in the operon that was predicted to be a T9SS component by the bioinformatic analysis. Furthermore, the interaction experiments showed that only PorD was associated with the T9SS and no other proteins expressed from this operon were involved. Besides, these other proteins are mostly of known function related to the central genetic processes of DNA replication and repair.

2. Methods are described in details therefore salt concentration used to wash beads charged with Sov antibodies and used for immunoprecipitation should be provided (lane 223).

This section in the method has been amended to include the 300 mM salt concentrations.

3. The results indicating PorW is required for Sov to form a high molecular weight complex are not convincing (Page 12) and should be verified by western blot analysis using anti-PorW antibodies. Alternatively cross-linking followed up SDS-PAGE and MS analysis should be considered.

We analysed almost all T9SS mutants (not all mutants are shown in the paper) and the high molecular weight Sov complex (720 kDa) was absent in the *porW* mutant only, providing convincing evidence that PorW is required for the high MW complex. Validation with anti-PorW would have been good, unfortunately our anti-PorW antibody did not detect PorW on BN-immunoblot. The role of PorW in binding to Sov was verified in both *P. gingivalis* and *E. coli* in our further experiments. Cross-linking followed by MS to identify the interacting domains is a good suggestion but is beyond the scope of this study.

4. Importance of PorD for the efficient operation of the T9SS needs to be verified qualitatively by comparing the activity of gingipains and PPAD, alternatively CPG70 secreted by WT-Pg and Δ porD.

We believe the immunoblot analysis we performed provided more insight as we were able to look at the A-LPS bound form as well as the unprocessed form. The ratio of unprocessed forms to A-LPS bound forms is a good indicator for defects/slowing in the processing which will be completely missed if we measured the activity of these proteins. Furthermore, the system might upregulate to compensate for less processed forms on the surface and so if the activity of these proteins were measured then we would see little difference. The major difference would be in the accumulation of the unprocessed form in the periplasm.

December 7, 2021

Prof. Eric C Reynolds
University of Melbourne
Oral Health CRC, Melbourne Dental School, Bio21 Institute
720 Swanston Street
Melbourne, Victoria 3010
Australia

Re: Spectrum01602-21R1 (Protein interactome analysis of the type IX secretion system identifies PorW as the missing link between the PorK/N ring complex and the Sov translocon)

Dear Prof. Eric C Reynolds:

Your manuscript has been accepted, and I am forwarding it to the ASM Journals Department for publication. You will be notified when your proofs are ready to be viewed.

Sincerely,

Cezar Khursigara
Editor, Microbiology Spectrum
